# A High-Precision Method for Segmentation and Recognition of Shopping Mall Plans

**DOI:** 10.3390/s22072510

**Published:** 2022-03-25

**Authors:** Ming Su, Wei Shi, Dangjun Zhao, Dongyang Cheng, Junchao Zhang

**Affiliations:** School of Aeronautics and Astronautics, Central South University, Changsha 410083, China; staylorm@csu.edu.cn (M.S.); zhao_dj@csu.edu.cn (D.Z.); cheng_dy@csu.edu.cn (D.C.); junchaozhang@csu.edu.cn (J.Z.)

**Keywords:** two-stage region growing, OCR, image segmentation, image recognition, shopping mall plans

## Abstract

Most studies on map segmentation and recognition are focused on architectural floor plans, while there are very few analyses of shopping mall plans. The objective of the work is to accurately segment and recognize the shopping mall plan, obtaining location and semantic information for each room via segmentation and recognition. This work can be used in other applications such as indoor robot navigation, building area and location analysis, and three-dimensional reconstruction. First, we identify and match the catalog of a mall floor plan to obtain matching text, and then we use the two-stage region growth method we proposed to segment the preprocessed floor plan. The room number is then obtained by sending each segmented room section to an OCR (optical character recognition) system for identification. Finally, the system retrieves the matching text to match the room number in order to obtain the room name, and outputs the needed room location and semantic information. It is considered a successful detection when a room region can be successfully segmented and identified. The proposed method is evaluated on a dataset including 1340 rooms. Experimental results show that the accuracy of room segmentation is 92.54%, and the accuracy of room recognition is 90.56%. The total detection accuracy is 83.81%.

## 1. Introduction

A floor plan is a graphical representation of the top view of a house or building along with its necessary dimensions. Relevant studies make use of a plan’s extensive architectural information to aid their own research [1,2,3]. For instance, ref. [1] focuses on detecting walls from a floor plan based on an alternative patch-based segmentation approach working at a pixel level, concluding that the identified walls can be utilized for a variety of tasks, including three-dimensional (3D) reconstruction and the construction of building boundaries. On the other hand, ref. [2] utilizes a two-dimensional (2D) floorplan to align panorama red-green-blue-depth (RGBD) scans, which can significantly reduce the number of necessary scans with the aid of a floorplan image. Finally, ref. [3] proposes a method for understanding hand-painted planar graphs by using subgraph isomorphism and Hough transform—acknowledging that a plan consists of recognizing building elements (doors, windows, walls, tables, etc.) and their topological properties—so as to propose an alternative computer aided design (CAD) system input technique that allows for the storage and modification of paper-based plans. Two-dimensional floor plan evaluation and information retrieval can help in many applications, e.g., to count the number of rooms and their areas, as well as architectural information recovery. Moreover, indoor robot navigation [4], indoor building area analysis, and position analysis [5] all require analyses of floor plans.

Furthermore, the technique for detecting rooms in an architectural floor plan has been investigated by various scholars [6,7,8,9,10,11,12]. For instance, a wall’s straight line is first detected based on an original coupling of classical Hough transform with image vectorization, and the door symbol is then discovered using the arc detection technique [13]; in a final step, the detected door and wall are connected to detect the room [9]. Ahmed et al. [10] adopted and extended this method [9] and introduced new processing steps such as wall edge extraction and boundary detection, and proposed a technique for automatically recognizing and labeling the rooms in a floor plan by combining the technology introduced in their own work [11].

Numerous studies have been performed on architectural floor plans, while research on shopping mall plans differs greatly from this field. To close the final minor gap in an architectural floor plan, ref. [12] employed speeded-up robust features (SURF) matching technology to recognize door symbols. This was not necessary in our research because a shop floor plan normally does not include an interval between sections, due to the door sign. That said, ref. [10]’s usage of room label information to assist with planar graph recognition is enlightening for our study. In a shopping mall plan, there are no sophisticated features, such as door symbols, to identify rooms, and there are numerous rooms on the plan that are tightly spaced and include a great deal of invalid information. As a result, the relevant approaches for an architectural floor plan are not applicable in shopping mall floor plans. Although a shopping mall plan does not have as much structural information as an architectural floor plan, it does include a great deal of semantic information: each room has a unique number that may be used to assist the study of the mall floor plan. Outdoor navigation technology has advanced dramatically in the last decade, but consumers are more likely to become disoriented in huge, enclosed structures such as stadiums, educational institutions, and shopping malls. Therefore, indoor plan analysis is crucial, but there has been no research on the segmentation and recognition of shopping mall plans. Therefore, this study focuses on two major issues: image segmentation and OCR.

Image segmentation plays a vital role in image processing. Image segmentation methods primarily include threshold, region, edge, and deep learning segmentation methods [14,15]. The key role of a threshold segmentation algorithm is to determine the threshold T for an image F, with 1 for target pixel F(i,j)>T and 0 for the non-target pixel F(i,j)<T; ref. [16] introduced a local dynamic threshold image segmentation method that overcomes the drawbacks of the global Otsu algorithm, removes the block effect, and improves the effectiveness and adaptivity of complex background image segmentation. To achieve image segmentation, the region growing method [17,18] first selects a seed pixel and then continuously merges pixels with similar characteristics. In order to overcome the influence of seed selection and segmentation holes, simple linear iterative cluster (SLIC) is used to segment the raw image into several small pixel blocks according to the characteristics of similar pixel gray levels [18]. The super pixel segmentation can eliminate the influence of image noise and uneven gray values. The edge-based segmentation method [19] mainly uses differential operators to detect the edge of gray mutations to achieve the purpose of segmentation. Typical differential operators, including the Roberts operator [20], the Sobel operator [21], the Canny operator [22], ref. [22] developed the Canny edge detection algorithm by using band decomposition and the Otsu algorithm to adaptively select high and low thresholds based on picture gray level, and performed well for remote sensing image segmentation.

The deep learning segmentation method [23,24] relies on a neural network to extract features, and outputs labels of each pixel end-to-end. Shelhamer et al. [24] proposed a full convolution neural network (FCN) semantic segmentation approach, which has demonstrated significant progress in this field. It not only solves the problem of how CNN achieves end-to-end semantic segmentation training, but it also effectively solves the problem of generating pixel level output semantic prediction for any size input. The two-stage region growth approach we propose is utilized to segment the plan to obtain each individual area to fulfill our objective in this article, i.e., the segmentation and recognition of each individual part of a mall plan. In addition, the preprocessing step employs threshold segmentation and edge detection technology.

OCR [25,26] takes in images containing text optical character information and outputs text information corresponding to those images. Currently, the mainstream OCR system is divided into two modules: text detection and text recognition. Recently, researchers have concentrated their efforts on developing techniques for digitizing handwritten documents, which are largely based on deep learning [27]. The use of cluster computing and GPUs, as well as the improved performance by deep learning architectures [28], such as recurrent neural networks (RNN), convolutional neural networks (CNN), and long short-term memory (LSTM) networks, has sparked this paradigm shift. Tencent Cloud [29] and Baidu Cloud [30], for example, have developed powerful and easy-to-use OCR technologies to allow user calls via application programming interfaces (APIs) to satisfy the needs of text extraction and analysis in production and real-life scenarios. Baidu OCR also offers multilingual recognition in a variety of scenarios with great robustness and accuracy. Furthermore, many academics have made common OCR projects open source, such as AdvancedEAST [31] (which is based on the Keras framework), PixelLink [32,33], and the connectionist text proposal network (CTPN) [34,35], which are implemented using the TensorFlow framework; EasyOCR [36], however, is built with the Pytorch framework. EasyOCR’s detection model employs the character-region awareness for text detection (CRAFT) network [37], while the identification model employs the convolutional recurrent neural network (CRNN) network [38]. EasyOCR has an advantage over other open-source programs in that it supports more than 80 languages, including English, Chinese (both simple and sophisticated), Arabic, Japanese, and other forms of identification. Due to language differences in the identification of catalogs and shopping mall plans in different countries, Baidu OCR and EasyOCR are primarily used as OCR identification modules in this paper to adapt to different language scenarios.

Each room number on the plan corresponds to an independent room region; although OCR can recognize the overall room numbers, it cannot judge which room number relates to which room region, nor can it determine the exact location of each room. We propose a method to segment a room by the two-stage region growth method, and then recognize the room by OCR to solve this problem. The remainder of this article is organized as follows. Section 2.1 describes the shopping center directory text matching system method. Section 2.2 describes the shopping mall room matching system’s processes in detail, including the preprocessing steps and the detailed content of the two-stage region growth algorithm, and introduces the use of an OCR system in the recognition module and region labeling. Section 3.1 outlines the experimental details and assessment criteria, followed by a large number of experiments to validate the algorithm’s effectiveness. Finally, discussion is provided in Section 4 and the conclusion is provided in Section 5.

## 2. Methods

This section is broken down into two parts to introduce (1) the Directory Text Matching System, which recognizes directories and obtains matched text, and (2) the Shopping Mall Matching System, which preprocesses and segments modules in detail; the recognition module is also discussed. Finally, the system preprocessing algorithm’s disadvantage is analyzed, and the multi-epoch detection approach is adopted to address the shortage.

Figure 1 depicts our system module. The Directory Text Matching System takes a directory image as input and returns the key value pair (matching text) of each row, with the room number as the key and the room name as the value. The matching text is fed into the Shopping Mall Plan Matching System to help the system finish the room matching operation. The input to the Shopping Mall Plan Matching System is a map corresponding to the directory. The preprocessed binary map is first obtained using the preprocessing module, and it is then segmented using the two-stage region growth approach to produce a region. The region is transferred to the OCR system to be identified and the room number is obtained. Then, the matched text is used to obtain the corresponding room name. Finally, the system generates a marked shopping mall plan, as well as information about the associated region, such as the corresponding room name and coordinate information.

### 2.1. The Directory Text Matching System

To begin, we recognize the plan’s directory and the OCR system returns the semantic information set Ω of all rooms after recognition (room number as the key and semantic information as the value), which is used as the input of the recognition module to assist in completing the matching work. Multiple columns of directory images need to be identified, and the position is usually uncertain. To obtain the text, we must first preprocess each column of the text image for identification and then transmit it to the OCR system. Algorithm 1 represents the implementation of directory text matching system.

**Algorithm 1** Directory text matching algorithm
**
*Input:*
**

$F_{0}(M\times N\times 3)$

***Output:*** Ω1: $p=selectCoordinate(F_{0})$
2: d=boundingRect(p)
3: M=getPerspectiveTransform(p,d)
4: $f(m\times n\times 3)=warpPerspective(F_{0},M)$
5: ***foreach***
f ***do***6:    line=getLinePts(f)
7: ***   for***
j∈f
***do***8: $\;\;\;\;\;\;f_{l}\leftarrow i<linepts$
9: $\;\;\;\;\;\;f_{r}\leftarrow i>linepts$
10: $\;\;text_{l},text_{r}\leftarrow f_{l},f_{r}\;do\;OCR$
11: $\;\;\Omega =match(text_{l},text_{r})$


#### 2.1.1. Preprocessing

The captured image may be slanted due to camera instability and inconsistent height between the camera and the picture, as shown in Figure 2a. To solve this problem, we used the perspective transformation algorithm [39] to correct the slanted picture; of course, this step could be skipped if there is no slant in the picture taken. Furthermore, if the room name in the directory is close to the room number, OCR will automatically recognize them as a string, as illustrated in Figure 2c. Thus, a line is drawn between the room name and the room number, and then f is divided into $f_{l}$ and $f_{r}$ photos based on this line, and they are submitted to OCR recognition, respectively.

#### 2.1.2. OCR Recognition and Matching

$f_{l}$ and $f_{r}$ are sent into the OCR system to properly detect and collect each string’s information (include meaning and location). EasyOCR only supports horizontal text, whereas Baidu OCR can recognize vertical text. Baidu OCR recognizes several vertical room numbers as a string in the experiment, which is inconvenient for text matching. As a result, we used EasyOCR to identify the directory, as illustrated in Figure 2d. Since the room name and number are typically in the same row in the catalog diagram, they can be matched based on the horizontal relationship of their corresponding coordinates. However, they cannot be entirely parallel, due to some flaws in human correction during the preprocessing stage. For matching, the method adopted is depicted in Figure 3. When $y_{r}$ in the right character’s center coordinate, satisfies $y_{r}\in \delta =\{y_{l}-\frac{h_{l}}{4},y_{l}+\frac{h_{l}}{4}\}$, it is considered the same row for matching. A key value pair is returned for each text row, in which the room number is the key and the room name is the value. To aid the mall plan matching system, the key value pair supplied by each row of all directories is set as a dictionary for the matching text.

### 2.2. The Shopping Mall Plan Matching System

Due to the influence of camera angles, lighting changes, complex backgrounds, and dense strings, the recognition accuracy of the entire map will be quite low if only OCR is used. Recognition accuracy may be substantially improved, and the precise location of each region can be determined, if each region is extracted separately and fed into OCR. Figure 4 depicts the Shopping Mall Plane Matching System.

#### 2.2.1. Preprocessing Module

To decrease the amount of invalid information in the shopping mall plan, a perspective correction transformation operation that is consistent with directory extraction is required to extract the map independently. Furthermore, to account for varying shooting angles and lighting conditions, the preprocessing module employs a mix of threshold segmentation, Canny edge detection, and exposure point detection.

To accelerate the region growing algorithm, $F_{g}$ is obtained by graying the shopping mall plan $F_{0}$, and the binary graph $F_{b1}$ is obtained by utilizing adaptive threshold segmentation for $F_{g}$. Each pixel is used as the center of the adaptive window Ω, and the pixel’s value is determined by calculating the threshold in Ω. The more pixels involved for calculation in Ω when T is bigger, the thicker the overall contour. However, the contour of $F_{b1}$ at the exposure point is broken due to the local focus of indoor lighting in a realistic photo. The local range of such breakpoints is considerable, as illustrated in Figure 5a, and it will cause excessive segmentation. Moreover, the exposure point $F_{g}(i,j)$ is relatively high. An appropriate threshold δ for threshold segmentation makes it easy to locate the exposure point, and the binary figure $F_{b2}$ can be obtained after the exposure point detection of $F_{g}$. In addition, as shown in Figure 5b, there are tiny breakpoints at some edges in $F_{g}$ due to the influence of light. To obtain the closed boundary graph, a hierarchy operation is performed on $F_{0}$ to separate the three channels $F_{R},F_{G},F_{B}$, and then a Canny edge detection operation is performed to obtain $F_{Rg},F_{Gg},F_{Bg}$. To obtain a closed border graph, $F_{b3}$, $F_{Rg}$, $F_{Gg}$, and $F_{Bg}$ are combined. Finally, the preprocessing result graph $F_{1}^{\prime }$ is created by combining $F_{b1}$, $F_{b2}$, and $F_{b3}$. After the above procedure, there are still some weak breakpoints. To obtain the final result $F_{1}$, the 3 × 3 rectangular symmetric structural element *S* is used to carry out the morphological dilation of $F_{1}^{\prime }$. Figure 5 depicts the preprocessing result.

#### 2.2.2. Segmentation Module

The preprocessed picture is segmented in the segmentation module using our proposed two-stage region growth algorithm, and the segmented region is then sent to the recognition module for matching. Algorithm 2 represents the implementation of two-stage region growth method.
**Algorithm 2** Two-stage region growing algorithm***Input:***$F_{1}(M\times N),F_{0}(M\times N\times 3)$***Output:*** $R_{2(RGB)},F_{growed}(M\times N)$1: $\left(i,j)=search\;initial\;seed\;(F_{1}\right)$
2: $(i,j)\;do\;region-growing\;to\;get\;R_{1}$
3: $R_{1}(m\times n)=\min box(R_{1})$
4: $\bar{R_{1}}(m\times n)=reverse(R_{1}(m\times n))$
5: $\bar{R_{1}}(m\times n)\;do\;edge\text{-}region-growing\;to\;get\;\bar{R_{2}}(m\times n)$
6: $R_{2}(m\times n)=reverse(\bar{R_{2}}(m\times n))$
7: $R_{2(RGB)}=AND(R_{2},F_{0})$
8: $R_{2(RGB)}\;to\;OCR\;systerm$


As shown in Figure 6a, after acquiring white region $R_{1}$ by region growing, the smallest rectangular frame $R_{1}(m\times n)$ of $R_{1}$ is taken and projected on $F_{0}(M\times N\times 3)$ to obtain $R_{1(RGB)}$. However, the pixels of the letter “C37” are not merged into $R_{1}$, resulting in the room region not being entirely segmented and OCR being unable to effectively recognize $R_{1(RGB)}$ to gain room semantic information. As a result, we proposed the two-stage region growing algorithm. As shown in Figure 6b, on the basis of obtaining $R_{1}(m\times n)$ by region growing, first reverse $R_{1}(m\times n)$ to get $\bar{R_{1}}(m\times n)$, then carry out region growing for the white region at the border of $\bar{R_{1}}(m\times n)$ to obtain the growth region $\bar{R_{2}}(m\times n)$, and finally reverse to obtain $R_{2}(m\times n)$ combining $R_{1}$ and internal character pixels; $R_{2}(m\times n)$ is projected onto $F_{0}(M\times N\times 3)$ to obtain $R_{2(RGB)}$, and $R_{2(RGB)}$ is then submitted to the OCR system for recognition. It was discovered that $R_{2}$ keeps the growing region $R_{1}$ while also combining internal character pixels, allowing the room area to be efficiently segmented and OCR to effectively recognize $R_{2(RGB)}$. On the basis of region growth, the two-stage region growing method retains the growing region $R_{1}$ and merges all the internal non-growing regions to generate the connected solid region $R_{2}$. We can successfully segment all of the effective regions in the room in meeting our objectives, and $R_{2(RGB)}$ is easier to identify than $R_{1(RGB)}$ in the recognition module to collect room semantic information.

Since the shopping mall plan includes a broad corridor, which is not our region of interest, when Formula (1) is met, it may be assumed that *R* is a corridor and not transmitted to the OCR system by judging the proportion of $R_{1}(m\times n)$ and $F_{1}(M\times N)$. When τ is 0.6, it can fulfill the accuracy criteria after numerous tests. Figure 7 depicts the corridor.
(1)$$\frac{m}{M}>\tau \;and\;\frac{n}{M}>\tau$$

#### 2.2.3. Recognition Module

We utilized two traditional OCR systems, EasyOCR and Baidu OCR, to recognize a room number and match it with the directory dictionary Ω, and eventually output $F_{2}(M\times N\times 3)$ labelled the room name and highlighted the pixel of the room region in the recognition module, which takes $R_{2(RGB)}$ obtained via two-stage region growing method as input.

Table 1 details the results of four kinds of $R_{2(RGB)}$. All the examples in Table 1 are identified by using Baidu OCR, highlighting $R_{2}{}^{1}$ as purple in $F_{0}$ if $R_{2(RGB)}{}^{1}$ is correctly recognized and matched. Due to an error in OCR induced by inadequate segmentation, $R_{2(RGB)}{}^{2}$ and $R_{2(RGB)}{}^{3}$ could not be successfully matched, since all pixels of $R_{2}$ can be obtained after two-stage region growing; $F_{0}$ is cropped to obtain $R_{3(RGB)}$ after traversing the pixels to obtain the maximum and minimum coordinates. If $R_{3(RGB)}$ can be successfully identified and matched, it is considered to be a valid target, and $R_{2}{}^{2},R_{2}{}^{3}$ is highlighted in red in $F_{0}$; otherwise, it is considered to be a non-target, and will not be marked. The highlighted $F_{0}$ serves as the recognition module’s output $F_{2}$. If there is no result after OCR, since the character is in the vertical direction, it should be rotated 90 degrees.

#### 2.2.4. Multi-Epoch Segmentation and Recognition Algorithm

The first three sections of this study detail the entire fulfillment of a shopping mall plan’s segmentation and recognition. However, if the character pixel is too close to the region boundary, they will be merged together, resulting in the OCR being unable to properly identify the character, as shown in 2 and 3 in Table 1. The fundamental cause is that, in the preprocessing module, the adaptive window Ω is relatively large and a dilation operation of $F_{1}^{\prime }$ is carried out, mostly to avoid overgrowth during two-stage region growing due to contour fracture. Such troublesome regions make up a small percentage of the total region to be identified in the map, but they account for the majority of the remaining undetected regions.

We used a multi-epoch segmentation and recognition strategy to solve this problem. The preprocessing techniques described in Section 2.2.1 were used in the first epoch, with T=25 in the adaptive window Ω and δ=230 in the exposure point detection. Morphological dilation was canceled in the second epoch, and T=5. Most crucially, $F_{\mathrm{detected}\text{\_}\mathrm{mask}}=1$ was set to record the detected region and corridor that were successfully marked in the first epoch; that is, $F_{\mathrm{detected}\text{\_}\mathrm{mask}}(i,j)=0$, where (i,j) belongs to the detected region and corridor, and then the 5 × 5 rectangular symmetric structural element S was utilized to perform morphological dilation on $F_{\mathrm{detected}\text{\_}\mathrm{mask}}$, which was joined with the preprocessing result $F_{2}^{\prime }$ to generate the final preprocessing result $F_{2}$, as illustrated in Figure 8.

Naturally, not all of the remaining undetected regions in the first epoch can be detected following the second epoch, so we developed a multi-epoch segmentation and recognition algorithm. Each epoch mixes the preprocessing module’s output $F_{i}^{\prime }$ with the previous epoch’s $F_{\mathrm{detected}\text{\_}\mathrm{mask}}{}^{i-1}$ as the segmentation module’s input $F_{i}$. The algorithm’s output is the sum of the result from each epoch. The flowchart of multi-epoch segmentation and the recognition algorithm is shown in Figure 9. Figure 10 depicts the experimental results.

## 3. Experimental Process and Results

This section initially provides the precision performance evaluation metric, as well as experimental data. Using that information, numerous experiments were carried out to confirm the algorithm’s effectiveness.

### 3.1. Introduction of Experimental Evaluation Criteria and Experimental Data

The accuracy performance assessment index has two components: segmentation precision and OCR precision. A detection is considered successful when a region can be successfully segmented and identified. As a result, the complete system’s accuracy index is based on detection precision. The following analysis provides a detailed definition of the evaluation index.

Table 2 serves as an illustration of the experimental evaluation criteria. In Formula (2), the total number of rooms in the image to be identified is represented by Ω. True positive (TP) denotes the number of regions that have been correctly segmented and recognized. $False\;Positive\;0\;(FP_{0})$ denotes the number of regions that are correctly segmented but result in matching mistakes as a result of recognition errors (affected by the accuracy of OCR system). $False\;Positive\;1\;(FP_{1})$ denotes the number of areas that are partly segmented and failed to recognize. False Negative (FN) denotes the number of areas that are not segmented successfully.
(2)$$\Omega =TP+FP_{0}+FP_{1}+FN$$

The precision performance evaluation metric for this work was determined using the statistical quantization results TP, $FP_{0}$, $FP_{1}$, and FN. In Equation (3), the SSR (the segmentation success rate, i.e., the proportion of successfully segmented regions to the total number of regions) is used to assess the two-stage region growing algorithm’s segmentation accuracy. The OCR accuracy is measured using the ISR (identification success rate, i.e., in the number of successfully segmented regions, the proportion of the number of successfully identified regions) in Equation (4). The whole performance of the proposed system is measured using the DSR (detection success rate, i.e., the proportion of successfully segmented and identified regions to the total number of regions, easy to determine as DSR=SSR×ISR), as stated in Equation (5). The precision performance evaluation metric of the whole system in this paper is mainly reflected by the calculated DSR.
(3)$$SSR=\frac{TP+FP_{0}+FP_{1}}{\Omega }$$
(4)$$ISR=\frac{TP}{TP+FP_{0}+FP_{1}}$$
(5)$$DSR=\frac{TP}{\Omega }$$

We used σ(second) to represent the entire time consumption of the segmentation and recognition algorithm. To minimize the effect of differing resolutions, we employed bilinear interpolation to scale the input image to approximate resolution. To maintain the original aspect ratio, the $h^{\prime }$ and the $w^{\prime }$ of the scaled image were determined using the following formula:(6)$$w^{\prime }=\frac{w}{\max(w,h)}\times 4096$$
(7)$$h^{\prime }=\frac{h}{\max(w,h)}\times 4096$$

Using Figure 11 as an example, we downloaded the appropriate mall plan from the official website of 25 retail malls as the dataset. This information can be downloaded at https://github.com/Staylorm13/shopping-mall-plans-dataset (accessed on 10 October 2021). In the statistical tables of the following three experimental results, Image 1 represents Figure 10a, including 44 rooms; Image 2 represents Figure 11, including 111 rooms; and the dataset represents the sum of 25 pictures, including 1340 rooms. The accuracy analysis of this algorithm is based on the experimental results of the dataset.

### 3.2. Experiment with Different OCR Systems

The conditions in which an OCR system is different, for example, Baidu OCR, can only be utilized by calling the appropriate API online. EasyOCR, unlike Baidu OCR, must utilize CUDA to identify locally without networking, and its recognition efficiency is dependent on system setup. Our experimental operating equipment was a personal computer with the following configurations: Windows 11 (8G RAM) with an NVIDIA GTX 1660s GPU and an Intel Core i5-10600 CPU.

In different scenarios, different OCR systems have variable recognition accuracy. We first performed OCR comparison tests to identify which OCR was used in the following studies in order to adapt to our application scenarios and select a superior OCR system. Table 3 shows the outcomes of the experiments utilizing various OCR technologies.

Table 3 shows that, when an OCR system is used alone, Baidu OCR’s recognition accuracy is 71.96%, higher than EasyOCR’s 61.96%, but the time consumption is also larger, since Baidu OCR must call the API online for recognition, and the speed is easily affected by network speed, whereas EasyOCR can be used locally by downloading the detection and recognition models in advance. The EasyOCR model is lighter than the Baidu OCR model; however, its recognition accuracy is lower. When we used EasyOCR and Baidu OCR simultaneously (the BOTH columns in the table show that both EasyOCR and Baidu OCR are used at the same time), the recognition accuracy was 83.64%, greater than that of using any OCR system alone, and the speed was equivalent to that of Baidu OCR. The complementary use of EasyOCR and Baidu OCR jointly improves the overall recognition accuracy. As a result, in the following tests, the OCR system integrating EasyOCR and Baidu OCR was employed as the recognition module. 

### 3.3. Experiment of Two-Stage Region Growing

We proposed a new approach in Section 2.2.2, i.e., the two-stage region growth method, which is based on region growth, keeping the region that has grown while merging all the internal regions that have not grown to produce solid regions. We could successfully segment all effective zones in the room in our task. Moreover, the segmented region created by two-stage region growth is simpler to use in recognizing and collecting room semantic information in the recognition module, improving OCR recognition accuracy. The effectiveness of the two-stage region growth algorithm in improving the accuracy of OCR in the recognition module was verified by using the region growth algorithm and the two-stage region growth algorithm, respectively, in the segmentation module. The comparative experimental results are shown in Table 4.

The experimental results show that the system’s overall recognition rate is just 5.4%, and its total detection rate is only 4.1% when the region growth method is utilized. The system’s recognition accuracy increased to 83.6%, with a total detection rate of 66.8%, after employing the two-stage region growth technique. The experimental results revealed that the two-stage region growth technique can considerably increase the system recognition module’s recognition accuracy and overall detection rate, and the pixels in each room can be obtained. The two-stage region growth technique is thus an essential component of map segmentation and recognition.

### 3.4. Experiment of Multi-Epoch Segmentation and Recognition

We discussed the influence of morphological dilatation operation and adaptive window size on image segmentation in Section 2.2.4. There will be a connection between string pixels and contour edges during the preprocessing stage, resulting in a decline in segmentation accuracy. As a result, we present a method of multi-epoch segmentation and recognition. The fundamental goal of this method is to improve the algorithm’s segmentation accuracy. Table 5, Table 6 and Table 7 illustrate the experimental results of the two-epoch algorithm in verifying the effectiveness of multi-epoch segmentation and recognition.

The purpose of Epoch Two is to make up for the system’s omission in Epoch One. On the basis of the experiment in Section 3.3, we continued the Epoch Two experiment; Table 5 and Table 6 are two-epoch experiments with region growing and two-stage region growing, respectively. The segmentation rate increased from 75.97% to 90.74% when Epoch Two of detection was performed, as shown in Table 5. Table 6 shows that, when only one epoch was performed, the segmentation accuracy was 79.85%, and the total detection accuracy was only 66.79%. When Epoch Two was performed, the system’s segmentation accuracy improved to 92.53%, and the total detection accuracy rose to 83.81%. To accelerate Epoch Two’s computational speed, the concept was carried out by integrating the detected regions recorded for Epoch One with the preprocessing results of Epoch Two. Table 6 shows that the average computational speed of Epoch Two was $\sigma_{2}=121\;s$, which is 10 s quicker than that of Epoch One, $\sigma_{1}=132\;s$. The speed of Epoch Two in Image 1 and Image 2 was nearly twice as fast as the speed in Epoch One; Epoch Two only needs to discover the remaining undetected regions after Epoch One has detected the majority of them. The experimental results demonstrated that multi-epoch segmentation and recognition can significantly improve the system’s segmentation accuracy and overall detection accuracy. Table 7 shows the experimental results of two-epoch segmentation and recognition utilizing two-stage region growing.

Figure 12 represents the experimental results of Figure 11, which correspond to the experimental data of Image 2 in Table 5, Table 6 and Table 7.

As shown in Figure 12a,b use the traditional region growth method, while Figure 12c,d use our two-stage region growth method. The room and the words are not connected when using traditional region growth, since the pixel values of words and room regions are not consistent, which is not convenient for OCR recognition, while our method solves this problem; Figure 12c,d has higher accuracy (more purple-filled rooms) compared with Figure 12a,b, which shows that our method is better than the traditional region growing algorithm. Furthermore, Figure 12a,c employ one-epoch detection, whereas Figure 12b,d employ two-epoch detection; Figure 12b,d has a higher segmentation accuracy (more rooms marked with color) than Figure 12a,c, indicating that the two-epoch detection method can improve segmentation accuracy.

## 4. Discussion

To validate the system’s efficacy, we ran a significant number of tests on the dataset including 1340 rooms. The segmentation accuracy was 92.54%, the recognition accuracy was 90.56%, and the overall detection rate was 83.81%. The two-stage region growing method we proposed can successfully segment all effective regions in the room and is favorable to OCR recognition, thus improving the overall detection accuracy. Our multi-epoch segmentation and recognition method effectively improves the system’s segmentation accuracy. Our system separates each room individually before identifying the segmented room and extracting the room’s semantic content. However, in a special case, where a room area is too tiny to display the room number (Figure 13a, “49” in Case 1), our system cannot properly segment and identify it, which is the main drawback of our approach in such a circumstance. However, after further investigation, we discovered that nearly all of the plan’s room numbers are included in the room area. Since Case 1 accounts for less than 5% of all rooms, our approach is effective in the majority of them. Furthermore, if the position of the room number and other irrelevant characters is close (Figure 13b, “C06” and “&” in Case 2), the OCR system will recognize the irrelevant characters and the room number as a string, and the “& C06” in Case 2 cannot be retrieved in the directory; thus, it cannot be successfully matched. In essence, this issue is still caused by OCR recognition errors, as it is easy for OCR to recognize characters that are close together. Case 2 may exist if two room numbers are present in the same room, although this only accounts for a minor fraction of the overall picture.

Finally, we presented the plan with the lowest precision in the dataset (Figure 13c, Case 3). The challenge of this plan is that it has numerous lines, many invalid regions, and the room number is contained in a circle, all of which make the OCR system’s recognition more difficult. Furthermore, the number’s location is quite close to the room’s edge, making room segmentation more difficult. Despite this, our algorithm detected 46 rooms (out of a total of 67) with a total accuracy of 68.65%. Our algorithm still needs to be developed to increase the detection accuracy of complex images and to adapt to more special circumstances (such as most numbers being outside the room) in order to tackle the challenges mentioned above. The recognition accuracy of the system’s OCR module has the largest effect on the overall system’s recognition accuracy. Follow-up research should examine better OCR technology to enhance the mall plan’s overall accuracy.

## 5. Conclusions

We proposed a comprehensive method for automated room segmentation and recognition, based on a shopping mall plan, to obtain the location and semantic information of each room, which can aid indoor robot navigation, building area and location analysis, and three-dimensional reconstruction. To extract and identify the room information, the system employs a number of structural and semantic analysis modules. The matching text is first collected from the directory. The mall plan map is then preprocessed, and the binary map is utilized for the two-stage region growing method to produce each segmented region, which information is then provided to the OCR system for identification to obtain the room number corresponding to the region. Finally, the matching text is retrieved to obtain the region information so as to obtain the high-precision automatic segmentation and recognition system. According to the results of the experiments, our method is capable of accurately segmenting and identifying each room; however, it does have some limitations. As a result, we will concentrate on algorithm improvement and detailed processing in the future to make the approach more practical.

## Figures and Tables

**Figure 1 sensors-22-02510-f001:**
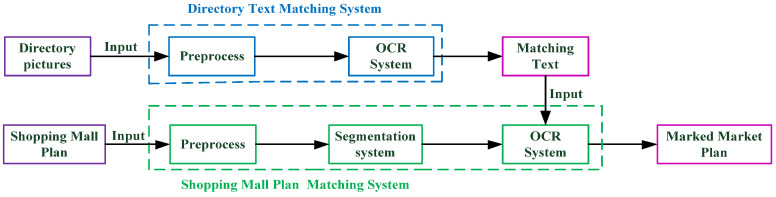
The entire procedure for segmenting and recognizing a shopping mall plan is presented. The Directory Text Matching System enters directory pictures to generate matching text, and the shopping mall plan is entered into the Shop Mall Plan Matching System to segment the region. Finally, the matched text is employed to aid recognition in order to acquire the information for each region.

**Figure 2 sensors-22-02510-f002:**
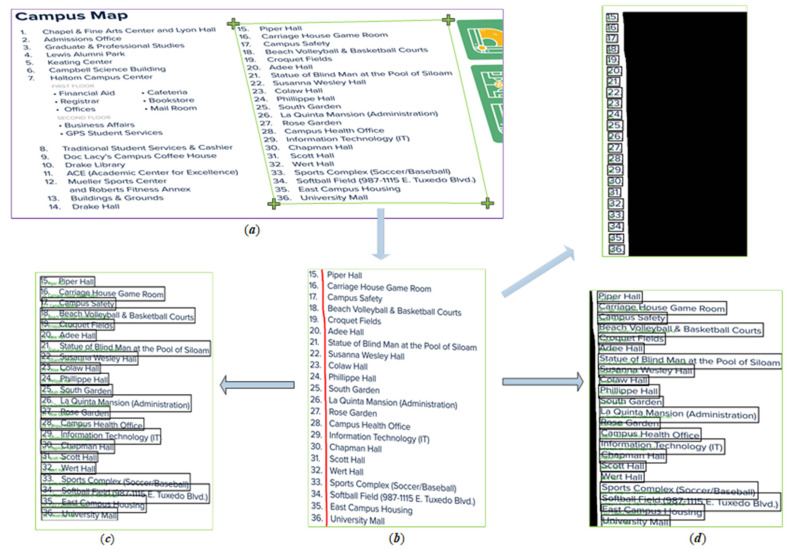
The procedure for the Directory Text Matching System is presented: (**a**) represents the original image, (**b**) represents the perspective correction results, (**c**) represents the unprocessed recognition results, and (**d**) represents the processed recognition results.

**Figure 3 sensors-22-02510-f003:**
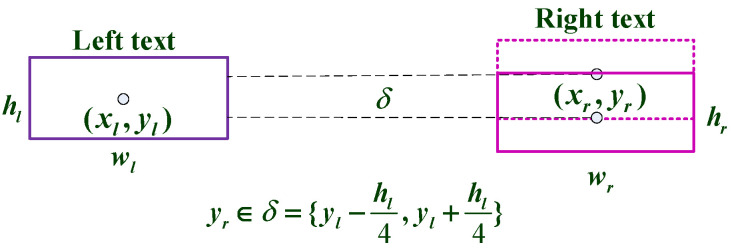
The two boxes represent the detected characters’ smallest rectangular box. Using the position of the left character as a guide, when the right character box confirms that $y_{r}\in \delta$, it is assumed that the two characters are roughly in the same horizontal position and can be matched.

**Figure 4 sensors-22-02510-f004:**
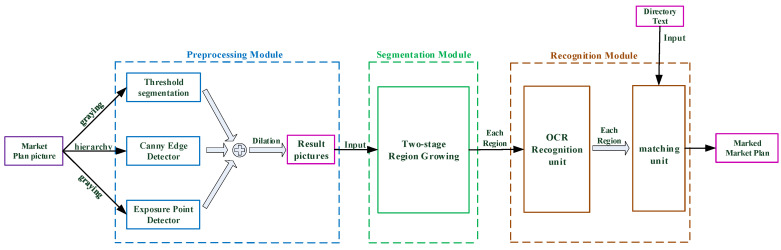
The entire procedure for the Shopping Mall Plan Matching System is presented. The preprocessing module uses the shopping mall plan as an input to generate a binary picture. In the segmentation module, the binary image is segmented to obtain each region to be identified. Each region is passed to the recognition module, in turn, to identify the room number, and the matching text is then traversed to acquire the corresponding region’s room name. After a successful match, the region is marked and the region’s information (room name and location) is returned.

**Figure 5 sensors-22-02510-f005:**
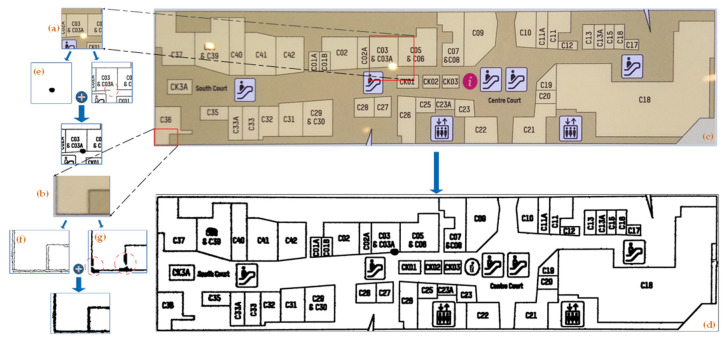
The preprocessing result is presented: (**a**,**b**) represent a partial enlarged view taken from $F_{0}$, (**c**) denotes the shopping mall plan $F_{0}$, (**d**) denotes the preprocessing result $F_{1}$, (**e**) denotes the result of exposure point detection $F_{b2}$, (**f**) denotes the result of canny edge detection $F_{b3}$, and (**g**) denotes the result of adaptive threshold segmentation $F_{b1}$.

**Figure 6 sensors-22-02510-f006:**
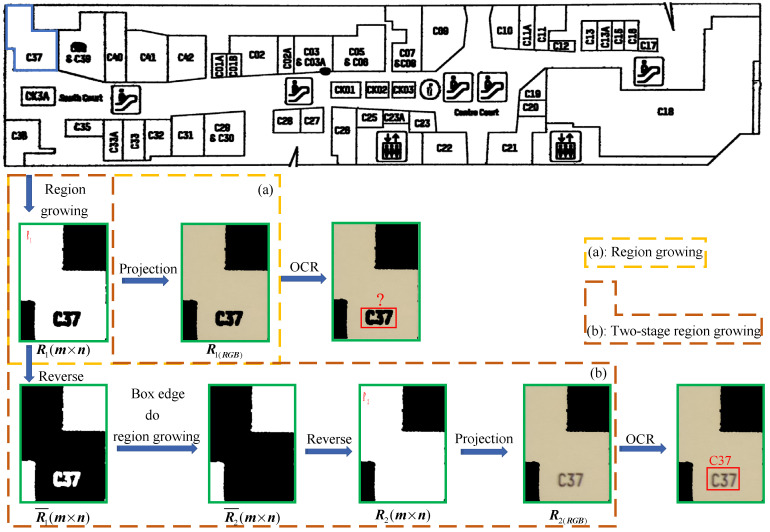
Schematic diagram of two-stage region growing algorithm, taking the “C37” region as an example. (**a**) represents the traditional region growing process, and (**b**) represents the two-stage region growing process.

**Figure 7 sensors-22-02510-f007:**
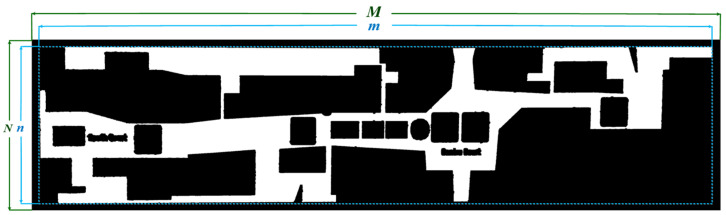
The white region denotes the corridor after region growing; *M*, *N* denote the size of $F_{1}$, and *m*, *n* denote the corridor’s minimum rectangle size.

**Figure 8 sensors-22-02510-f008:**
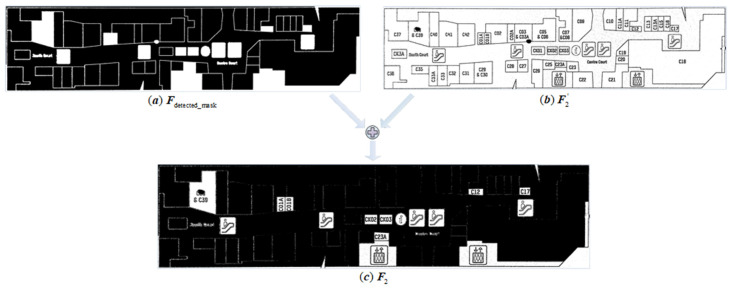
(**a**) represents Epoch One’s detected results $F_{\mathrm{detected}\text{\_}\mathrm{mask}}$, (**b**) represents Epoch Two’s preprocessing results $F_{2}^{\prime }$, and (**c**) represents the segmentation module’s input $F_{2}$ of Epoch Two.

**Figure 9 sensors-22-02510-f009:**
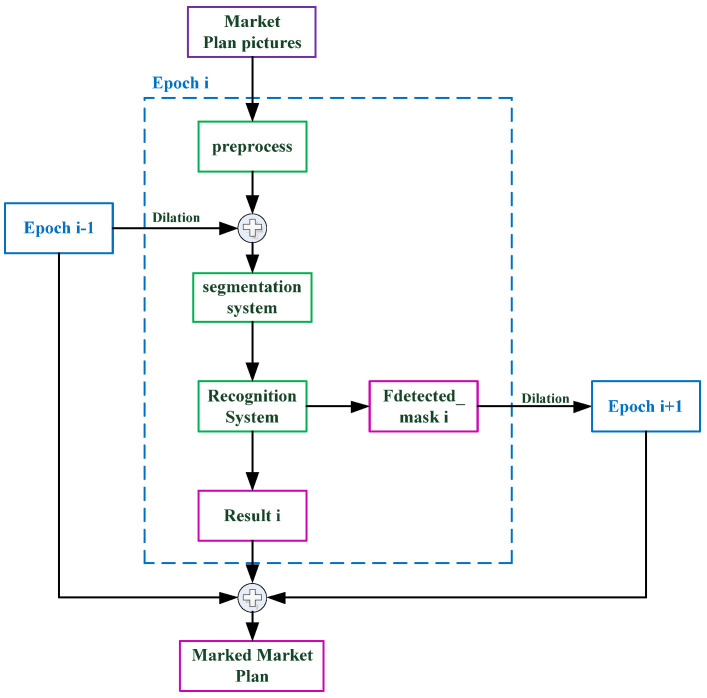
The flow chart of multi-epoch segmentation and recognition is shown.

**Figure 10 sensors-22-02510-f010:**
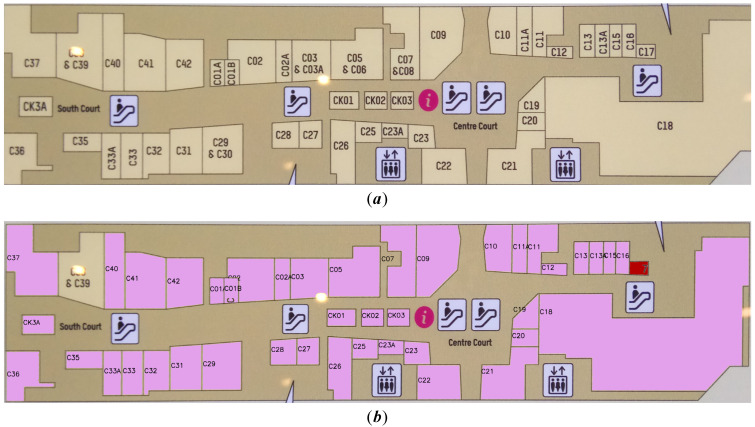
(**a**) representation of the raw picture, and (**b**) representation of the outcomes of segmentation and recognition, following two epochs of segmentation and recognition. The regions of successful segmentation and recognition is filled with purple and labeled with the associated room number, whereas the successful segmentation (incomplete segmentation) but incorrect recognition area is labeled with red.

**Figure 11 sensors-22-02510-f011:**
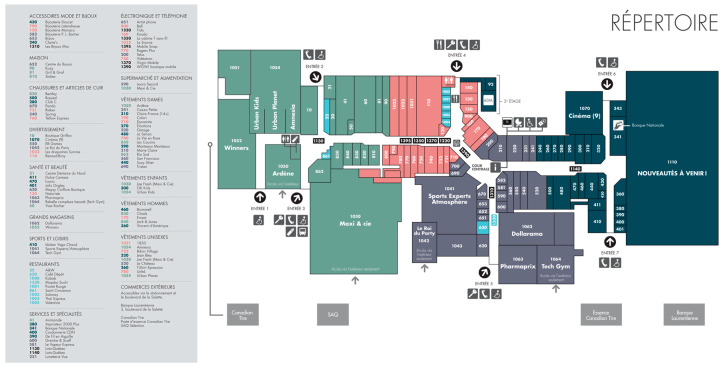
Shopping mall plan. This can be downloaded at https://www.carrefourdunord.com/en/informations/sitemap (accessed on 10 October 2021).

**Figure 12 sensors-22-02510-f012:**
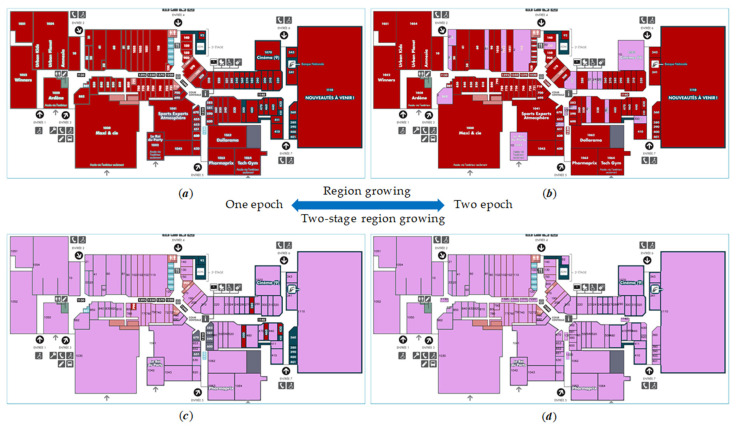
(**a**) represents the one-epoch result with traditional region growing; (**b**) represents the two-epoch result with traditional region growing; (**c**) represents the one-epoch result with two-stage region growing; (**d**) represents the two-epoch result with two-stage region growing. The purple-filled region denotes a room that has been successfully segmented and identified, whereas the red-filled area denotes a room that has been successfully segmented but incorrectly recognized. Rooms that are not successfully segmented still retain the colors in the original picture (without red and purple filling).

**Figure 13 sensors-22-02510-f013:**
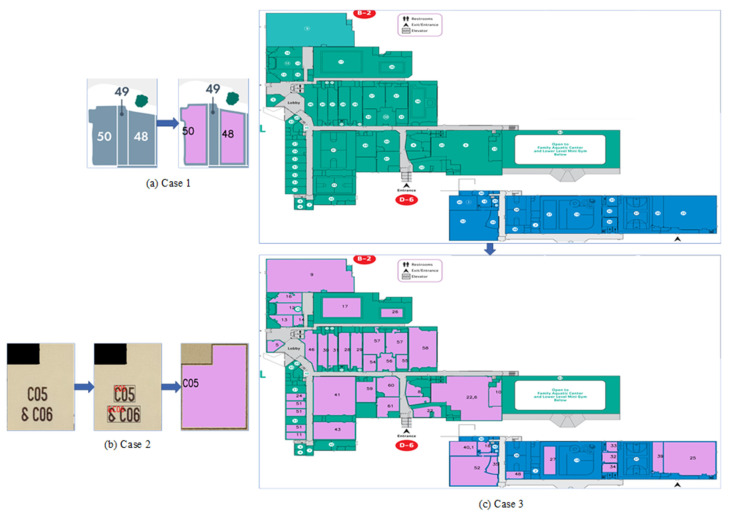
Special case analysis. (**a**) represents the room number outside the room. (**b**) represents that the room number that is connected with irrelevant characters. (**c**) represents the plan with the lowest accuracy in the dataset, and the purple area represents the rooms successfully segmented and identified.

**Table 1 sensors-22-02510-t001:** Results of four input types of recognition module.

	$R_{1}(m\times n)$	$R_{2}(m\times n)$	$R_{2(RGB)}$	Target	Segmentation	Recognition	$R_{3(RGB)}$	Recognition	$F_{2}(R_{2})$
1	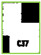	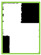	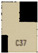	√	√	C37: cafes	None	None	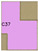
2	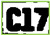	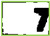	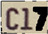	√	Partly	C1: ×	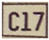	C17: dress shops	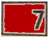
3	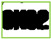	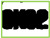	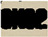	√	Partly	×	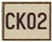	CK02: EBGames	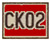
4	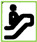	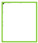	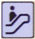	×	√	×	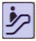	None	None

**Table 2 sensors-22-02510-t002:** Example of evaluation index.

	TP	$FP_{0}$	$FP_{1}$	FN
Test samples	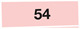	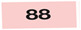	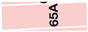	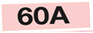
Test results	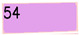	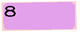	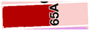	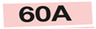

**Table 3 sensors-22-02510-t003:** Experiment result with different OCR systems.

Image	EasyOCR			Baidu OCR			BOTH		
	DSR	ISR	σ(second)	DSR	ISR	σ(second)	DSR	ISR	σ(second)
1	0.613636	0.627907	72 s	0.795455	0.813953	124 s	0.818182	0.837209	114 s
2	0.675676	0.937500	111 s	0.675676	0.937500	188 s	0.675676	0.937500	179 s
Dataset	0.494776	0.619626	92 s	0.574627	0.719626	127 s	0.667910	0.836449	132 s

**Table 4 sensors-22-02510-t004:** The comparative experimental results.

Image	Region Growing				Two-Stage Region Growing			
	DSR	SSR	ISR	σ(second)	DSR	SSR	ISR	σ(second)
1	0.000000	0.954545	0.000000	106 s	0.818182	0.977273	0.837209	114 s
2	0.000000	0.801802	0.000000	174 s	0.675676	0.720721	0.937500	179 s
Dataset	0.041045	0.759701	0.054028	125 s	0.667910	0.798507	0.836449	132 s

**Table 5 sensors-22-02510-t005:** Experiment result with traditional region growing.

Image	Epoch One				Epoch Two			
	DSR	SSR	ISR	$\sigma_{1}^{\prime }(\mathrm{second})$	DSR	SSR	ISR	$\sigma_{2}^{\prime }(\mathrm{second})$
1	0.000000	0.954545	0.000000	106 s	0.568182	0.977273	0.581395	105 s
2	0.000000	0.801802	0.000000	174 s	0.036036	0.981982	0.036697	221 s
Dataset	0.041045	0.759701	0.054028	125 s	0.220149	0.907463	0.242599	140 s

**Table 6 sensors-22-02510-t006:** Experiment result with two-stage region growing.

Image	Epoch One				Epoch Two			
	DSR	SSR	ISR	$\sigma_{1}(\mathrm{second})$	DSR	SSR	ISR	$\sigma_{2}(\mathrm{second})$
1	0.818182	0.977273	0.837209	114 s	0.954545	0.977273	0.976744	61 s
2	0.675676	0.720721	0.937500	179 s	0.936973	0.936973	1.000000	87 s
Dataset	0.667910	0.798507	0.836449	132 s	0.838060	0.925373	0.905645	121 s

**Table 7 sensors-22-02510-t007:** Experiment result detail of Epoch Two with two-stage region growing.

Image	TP	$FP_{0}$	$FP_{1}$	FN	Ω
1	42	0	1	1	44
2	104	0	0	7	111
Dataset	1123	18	99	100	1340

## Data Availability

Not applicable.

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
