# Peer review of "A High-Precision Method for Segmentation and Recognition of Shopping Mall Plans"

_sensors, 2022, doi:10.3390/s22072510_

Round 1
Reviewer 1 Report
This paper proposes a high-precision method for the segmentation and recognition of shopping mall plans. The proposed method aims to obtain the location and the semantic information of each room using a region-based segmentation algorithm combined with an OCR system. However, some parts of the paper seem to require additional work. The main issues are as follows:
- In Section 2.2.3, the Authors mentioned the traditional OCR systems used and put some results in Table 1. However, the author did not mention which OCR system is used in the examples. Did all the examples in Table 1 use the same OCR systems? The author should explain it.
- In Table 2, the Authors put the example of evaluation index on Raw and label. What does the Raw and label mean? Is the label represent the ground-truth? The authors should use standard computer vision term to avoid misunderstanding.
- Authors should explain Figure 12 in more detail. What is the difference between the two pictures? In addition, why do they have a different color?
- Authors should do thorough proofreading.
Author Response
Dear reviewer:
Thanks you for your valuable comments. We have revised the manuscript according to your valuable advice.
Please review Response 1.docx and sensors-1630366-revised.docx , Thank you.
Yours sincerely
Wei Shi

Reviewer 2 Report
Some words are colored red, the reason for which is unknown.
The summary is not clear regarding the objectives of the work, as it starts by introducing the shopping mall plans and then details the specific case of the rooms.
The state of the art presented in Chapter 1 is complete but too summarized. The authors could have included a little more information about each paper analyzed.
The authors start by doing OCR to identify the number of rooms, but it makes more sense that this task is the second task because the rooms are not yet identified.
As shown in figure 2a, room captions can be written on an image that appears slanted, but why? The person taking pictures can get a rectangular, non-deformed image.
Algorithm 2 has no relevant information, line 3 of the algorithm is confused. This algorithm can be removed, as the main steps are already described in the text.
Although not explicitly mentioned in the text, an alternative to growing regions is to use a threshold in the histogram of the equalized image, which could generate results faster. Why wasn't it used?
The image processing, as shown in Figure 5 (1), assumes that the room numbering is inside the room. But when this does not happen and the numbering is outside the room, how does the algorithm behave?
The authors do not need to be presenting all the algorithms used, because many of them implement the methods known to everyone, for example: growing regions. Authors should present algorithms only when it is something new or complex to be explained in words.
Table 1 is much more useful than the algorithms, as it condenses the information. The authors should re-think whether it is very important to include the algorithms.
Comparing figures 10 (a) and (b) it can be seen that C03A, C06 and C08 in (a) do not appear as text in (b). A justification must be included.
The authors claim that they analyzed 25 images in a total of 1340 rooms. Figure 10 shows one of the images and the result of the identification of the rooms. In figure 11 they present an image before being processed and in figure 12 the result. Considering that regardless of the value of the metrics, analysis by visual inspection is very important, it is recommended that authors include 1 or 2 more cases of images that they have analyzed and the respective result, one of these cases being the "worst case".
Author Response
Dear reviewer:
Thanks you for your valuable comments. We have revised the manuscript according to your valuable advice.
Please review Response 2.docx and sensors-1630366-revised.docx , Thank you.
Yours sincerely
Wei Shi

Round 2
Reviewer 1 Report
The authors have satisfactorily answered my questions.
This manuscript is a resubmission of an earlier submission. The following is a list of the peer review reports and author responses from that submission.
Round 1
Reviewer 1 Report
A high-precision method for segmentation and recognition of shopping mall plans
The authors are proposing a multi-epoch algorithm that employs several structural and semantic analysis modules for the segmentation and recognition of shopping mall plans. The research is thrilling with general applications in indoor robot navigation, indoor building area, and position analysis. However, I have identified a few issues to be addressed in the manuscript before making a recommendation for publication.
- The language of the manuscript is generally good. However, some undefined acronyms such as OCR, CAD, RGBD, APIs were introduced without being defined where first used. Moreover, there are a few typos such as a missing space between the phrases “[28,29]takes” on line 82, “p.The” on line 156, and the phrase “… can be get after …” on line 216 should be replaced by “… can be obtained after …”.
- Missing references in “… region growing [¬]” and “… and merging [¬]” on lines 69 and 70 respectively.
- The statements from line 106 to line 121 and Figure 1 should be moved to Section 2 at the beginning of line 142. They are more appropriate in the method section than the introductory section.
- Texts on lines 150 and 163 should be numbered as “2.1.1 Preprocessing” and “2.1.2 OCR recognition and matching” respectively.
- The authors claimed in Section 2.2.2 that randomization or manual selection is a technique for initial seed in the region growing algorithm without reference support. One of the conventional methods for seed selection and growth strategy in the adaptive seeded region growing is based on edge detection, texture extraction, and cloud model. How is the initial seed method of this study compared to this conventional method?
- The experimental image data set and evaluation metrics should be described in Section 3.1. It is not clear what is meant by “3.1 Details”, and how the experimental images were acquired should be described.
- The precision performance evaluation metric has not been used in the manuscript to measure the precision of the proposed algorithm, so what makes the algorithm high precision reflected in the title of the manuscript?
Author Response
Thanks for the opportunity to revise our manuscript. These comments are all very valuable to improve our paper. We have thoughtfully considered these comments and made corrections accordingly. Please see the attachment.

Reviewer 2 Report
I think this article has to accept in present form.
Author Response
Thank you very much for your affirmation. We will continue to make efforts in academic research.
Reviewer 3 Report
The manuscript presents an image processing and analysis application that combines (almost) monochromatic object segmentation and OCR to read mall plan.
Specific remarks
title:
'High-precision' is unclear in the title, is it relative to the position precision, the OCR precision ?
abstract:
Figures given in the abstract relative to the performances achieved should be put into context, what is a considered as good detection ?
l.24-212
The previous work session is rather long and not specific enough to the tackled problem.
l.106
Despite the fact that OCR can recognize all input content, it is unable to distinguish between content independence.
Please explain more what is independence in this context.
l.463
Objective is unclear, is it meant to be deployed real-time on a phone, 365 sec is obtained on what type of hardware ?
general remark
The novelty of the paper is unclear.
The abstract claims 'The proposed method is evaluated on abundant testing data source' it is unclear what data source is utilized ?
Lack of generalization, only 2 images have been tested, plus eventually one 'ablation' test which is not very well described.
Author Response

(The authors gave the same response as above.)

Reviewer 4 Report
This paper proposes a high-precision method for the segmentation and recognition of shopping mall plans. The proposed method aims to obtain the location and the semantic information of each room using a region-based segmentation algorithm combined with an OCR system. However, some parts of the paper seem to require additional work. The main issues are as follows:
- The major problem of this paper is that the authors just combine existing technique (like OCR, edge detection, region grow …etc.) in order to achieve a specific goal in the field of computer vision. Authors should at least propose a new method for the target problem. If they only used the existed method, then this paper has no novelty at all and it should not be considered as a journal paper.
- Authors should put the results, which explain in lines 116-118 into the abstract for better representation.
- In Section 1, lines 125-126 Authors mentioned the organization of the paper, which the conclusion part is in Section 4. However, in the manuscript, Section 4 is the discussion. Authors should write the correct organization.
- In Section 2.2.2, lines 285-292 explain the post-processing of the failed cases. The authors explain that the OCR fails to properly identify the ‘C37’. How to determine the important region since this process is not the part of recognition module (which utilizes the OCR methods)?
- The authors should put more examples of the results of the experiment. For example, in Section 3.4 experiments of multi-epoch, since the previous section explains the ablation study on post-processing technique, Authors should conduct the experiments on it.
Round 2
Reviewer 3 Report
As mentioned before, the added value to the current state of the art is marginal, the application is rather limited and the validation/test is done on a very limited dataset (no change since the first submission).
Therefore the reject decision.